# Environmental health literacy competencies and teaching methods for students and health professionals in Primary Health Care: A scoping review

Flaviane Cristina Rocha Cesar[1]*, Fabian Calixto Fraiz[2], Aline Maciel Monteiro[3],
Maria Alves Barbosa[4], Ricardo Cambraia Parreira[1], Andréa Cristina de Sousa[1],
João Victor Pereira Almeida[1], Jaqueline Reis Sotero da Silva[5]

1 School of Medicine, Centro Universitário de Mineiros (UNIFIMES), Trindade, Goiás, Brazil, 2 School of Dentistry, Universidade Federal do Paraná (UFPR), Curitiba, Paraná, Brazil, 3 School of Medicine, Universidade de Rio Verde (UniRV), Aparecida de Goiânia, Goiás, Brazil, 4 Graduate Program in Health Sciences, Universidade Federal de Goiás (UFG), Goiânia, Goiás, Brazil, 5 School of Nursing, Centro Universitário Sul-Americano, Goiânia, Goiás, Brazil

☉ These authors contributed equally to this work.
* flaviane.rocha@unifimes.edu.br

## Abstract

Environmental determinants play an increasingly important role in shaping population health, highlighting Environmental Health Literacy (EHL) as a critical competency for Primary Health Care (PHC). Despite growing attention to environmental health education, evidence addressing EHL in PHC remains fragmented and inconsistently conceptualized. This scoping review aimed to systematically map existing evidence on environmental health literacy among health professions students and practicing professionals in PHC settings, with a focus on identifying core competency domains and educational interventions relevant to PHC practice. A scoping review was conducted in accordance with PRISMA-ScR guidelines. Searches were performed across seven electronic databases, complemented by additional searches in Google Scholar and grey literature sources. Empirical and theoretical studies examining EHL or closely related environmental health constructs in PHC or PHC-relevant contexts were included. Twenty-nine studies met the inclusion criteria, comprising 24 empirical studies and five theoretical contributions. This review represents the first synthesis explicitly framed within the EHL context for PHC. Across studies, five core EHL domains consistently emerged: environmental health knowledge, clinical skills, risk communication, equity-oriented attitudes, and systems-based advocacy. Building on this synthesis, the review proposes a consolidated set of global competencies for EHL in PHC and systematically maps educational and practice-based interventions aimed at developing these competencies, particularly multimodal and technology-enhanced approaches. While several interventions demonstrated improvements in knowledge, skills, and self-efficacy, evidence of sustained integration into routine PHC practice

**Data availability statement:** All data underlying the findings of this study are publicly available in the following figshare repository: https://doi.org/10.6084/m9.figshare.30755477.

**Funding:** The Centro Universitário de Mineiros (UNIFIMES) provided financial support to cover the article processing charge (APC) through reimbursement of the publication fee. No additional external funding was received for this study. The funders had no role in study design, data collection and analysis, decision to publish, or preparation of the manuscript.

**Competing interests:** The authors have declared that no competing interests exist.

remained limited. Overall, the findings demonstrate that EHL is a multidimensional and foundational competency for PHC but is most often addressed implicitly rather than explicitly. Bridging the gap between education and clinical application will require sustained curricular integration, institutional support, and system-level alignment to ensure that PHC professionals are equipped to address environmental determinants of health in an increasingly complex global context.

## Introduction

The contemporary environmental crisis poses an unprecedented challenge to global health, driven by interconnected disruptions such as pollution, climate change, and biodiversity loss that amplify disease burdens and exacerbate inequities worldwide [1]. The World Health Organization estimates that a substantial proportion of global mortality is attributable to modifiable environmental risk factors, disproportionately affecting vulnerable populations, particularly children in low- and middle-income countries (LMICs) [2]. Primary Health Care (PHC), situated at the frontline of population health, is a critical approach for identifying, preventing, and mitigating environment-related health risks, yet remains insufficiently prepared to address the complex interplay between environmental exposures and social vulnerabilities [3].

PHC professionals routinely encounter conditions associated with environmental degradation, such as respiratory illnesses linked to air pollution, vector-borne diseases influenced by climatic shifts, and chronic conditions aggravated by toxic exposures, yet they face persistent barriers to integrating environmental considerations into clinical practice [1]. These include overcrowded curricula that marginalize environmental health, limited faculty expertise, insufficient tools for environmental exposure assessment, and scarce patient education resources [3,4]. PHC settings that care for populations experiencing high social vulnerability are particularly affected, as environmental injustices compound pre-existing health inequities [2].

Environmental Health Literacy (EHL) has emerged as a key competency domain for strengthening PHC responsiveness to environmental threats. In this review, EHL is defined as the integrated set of knowledge, clinical skills, risk communication abilities, and community-engagement capacities that enable health professionals to identify, assess, and respond to environmental determinants of health. For PHC practitioners, this includes obtaining environmental exposure histories, offering anticipatory environmental guidance, interpreting environmental health information, supporting patients and communities in reducing risks, and advocating within broader public health systems [3,5]. Given their longitudinal and population-wide engagement, PHC professionals serve as trusted messengers whose EHL can influence individual behaviors, inform local policy, and contribute to community resilience [6].

The importance of EHL for PHC is reinforced by the unique role of primary care providers at the interface between clinical care and community health. They are positioned to detect environmental exposures early, monitor related outcomes over

time, and integrate environmental risk assessment into routine preventive care [2]. Participatory and community-engaged approaches to environmental health education, which center on co-learning, recognition of local knowledge, and collective action, align closely with PHC principles [7].

While isolated curricular innovations demonstrate feasibility, such as train-the-trainer programs that prompted residency-level integration of environmental content [8], systematic and sustained curricular incorporation remains rare. Nursing and medical programs frequently report insufficient environmental health content, limited faculty preparedness, and competing priorities that marginalize this domain [1,4]. Competency frameworks vary widely, from pediatric environmental health [2] to community-engaged models [7], but few address the specific competencies required for PHC practice. Evidence related to competency assessment, implementation strategies, and applicability to resource-constrained environments remains limited [9,10].

Recent scoping and systematic reviews on planetary health and environmental health education demonstrate progress while highlighting persistent gaps. A global scoping review identified 73 planetary health education articles but noted that 88% originated in high-income countries and few addressed PHC-specific needs [11]. Another review of planetary health in medical education found that most publications (71%) focused on calls to integrate content rather than documenting implementation strategies or educational outcomes [12]. Reviews in nursing education reveal similar patterns, with a predominance of commentaries and limited empirical evidence or interprofessional approaches [13,14]. Remarkably, no completed scoping or systematic review has synthesized evidence on EHL within PHC settings, especially in LMIC contexts. A registered protocol for a scoping review on planetary health in family medicine continuing professional development indicates emerging interest but remains unpublished [14].

Thus, the existing review literature highlights two gaps: geographic concentration in high-income settings and a thematic emphasis on medical and nursing education in general. There is insufficient attention to the distinctive competencies, frameworks, and pedagogical strategies required to build EHL among PHC students and professionals who deliver frontline care in diverse global contexts. Given the conceptual diversity, methodological heterogeneity, and fragmented nature of this literature, a scoping review is the most appropriate method to systematically map available evidence, clarify existing frameworks, and identify gaps. [15].

By mapping the conceptual frameworks, competencies, content domains, and teaching methods used to develop EHL for PHC students and professionals, this review will provide a structured evidence base for educators, curriculum developers, and policymakers. It may inform context-sensitive curricular design, faculty development initiatives, and national or international standards for environmental health integration [1]. In LMIC contexts, where environmental burdens are greater and documentation of educational strategies is scarce, the findings may guide the development of resources, capacity building, and South-South collaboration [9,10].

## Review question

What are the conceptual frameworks, competencies, content areas, and teaching methods currently used in the development of environmental health literacy for students and health professionals in Primary Health Care, and how are these elements applied in different educational settings?

## Objectives

This scoping review aims to systematically map existing evidence on environmental health literacy for students and professionals in PHC settings, specifically to identify and synthesize:

(1) the core competencies and learning outcomes proposed or implemented for PHC trainees and practitioners; and

(2) the teaching strategies, pedagogical methods, and educational interventions employed to develop EHL in PHC contexts.

## Method

A scoping review of the literature was employed to examine the specificities and complexities of the concept of environmental health literacy, particularly in the education and practice of health professionals within the context of primary care. This methodological approach is especially suitable for synthesizing and mapping emerging or heterogeneous fields, where conducting a systematic review may be unfeasible or premature. Scoping reviews play a critical role in clarifying concepts, identifying key elements within the existing literature, and uncovering knowledge gaps that warrant further investigation [16]. This approach aligns with the objectives of the present study, which aims to comprehensively map the current state of environmental health education in primary care in order to support the development of a robust and context-sensitive training framework. The study protocol was registered in the Open Science Framework database (registration number: osf.io/8u7wd, October 2025).

### Search strategy

The literature search was conducted in accordance with the three-step strategy recommended by the Joanna Briggs Institute (JBI) for scoping reviews [16], ensuring methodological rigor and comprehensive coverage of the topic:

1. **Initial Search**: A preliminary and limited search was performed in MEDLINE (via PubMed) and Web of Science to identify relevant descriptors and indexing terms associated with the concepts of *environmental health literacy*, *health competencies*, and *primary health care*. Titles, abstracts, and indexing terms of the retrieved studies were examined to refine the search vocabulary.

2. **Comprehensive Search**: Based on the terminology identified in the initial stage, a second and more extensive search was conducted across multiple databases, including MEDLINE (via PubMed), Embase, Scopus, Web of Science, CINAHL, and LILACS. For each database, the search strategy was customized using controlled vocabularies (e.g., MeSH, Emtree) and relevant synonyms. The complete search strategies for each source are detailed in Appendix 1.

3. **Search for Additional Sources**: To ensure the inclusion of potentially relevant studies not captured in the database searches, the reference lists of all selected articles were manually screened. Additionally, grey literature was explored through platforms such as OpenGrey, Google Scholar, ProQuest Dissertations & Theses, and various institutional repositories.

No restrictions regarding language or publication date were imposed. However, only studies published in English, Portuguese, or Spanish were included for full-text screening.

### Study selection

This scoping review encompassed studies that fulfilled specific eligibility criteria, defined according to the Population–Concept–Context (PCC) framework recommended by the JBI. The aim was to map existing evidence on the EHL competencies of health professionals within PHC settings.

**Population:** The review included studies focusing on health professionals active in PHC, such as physicians, nurses, dentists, social workers, community health workers, and other allied health practitioners. Research involving health professionals in training, such as undergraduate students and medical or residency trainees, was also considered provided it addressed aspects related to environmental health literacy competencies.

**Concept:** The central concept guiding inclusion was the development and application of environmental health literacy competencies, which are understood as a combination of knowledge, skills, and attitudes.

- *Knowledge* encompassed cognitive dimensions involving the understanding of environmental determinants of health, including issues such as pollution, climate change, and environmental exposures.

- *Skills* referred to the ability to apply environmental health knowledge in practice, particularly within clinical or community contexts, involving activities such as risk assessment, health communication, and multisectoral coordination.

- *Attitudes* addressed the values, beliefs, and perceptions of health professionals regarding the integration of environmental health into their routine care practices.

Additionally, studies describing educational, clinical, or organizational interventions aimed at enhancing EHL responsiveness in academic or professional settings were also included.

**Context:** Eligible studies were situated in primary health care contexts, regardless of geographic location, and included both urban and rural areas.

**Types of Sources:** A wide range of evidence was considered, including empirical studies (quantitative, qualitative, and mixed methods), institutional reports, curricular documents, training guidelines, and theoretical frameworks related to environmental health competencies.

**Language and Timeframe:** No temporal restrictions were applied to ensure comprehensive inclusion of relevant literature. Publications in English, Portuguese, and Spanish were considered. The literature search was concluded in November 2025.

**Exclusion Criteria:** Studies were excluded if they did not focus on EHL-related competencies, if they were conducted in secondary or tertiary care settings without direct relevance to PHC, or if they consisted solely of opinion articles, editorials, or narrative reviews lacking empirical or conceptual grounding.

To ensure methodological transparency, all records were managed using *Rayyan*, a software platform specialized for systematic reviews. Duplicate entries were removed, and titles and abstracts were independently screened by two reviewers according to the established inclusion and exclusion criteria. Full texts of potentially relevant studies were retrieved and assessed independently by the same reviewers. Discrepancies were resolved through discussion or consultation with a third reviewer when necessary. As is consistent with scoping review methodology, no formal critical appraisal of the included studies was undertaken.

## Data extraction

Data from the selected studies were systematically extracted using a standardized data charting form, specifically adapted to align with the objectives of this scoping review. The form was designed to capture key information relevant to the research questions and included the following elements: (1) Author(s), (2) Year of publication, (3) Country of origin, (4) Study population, (5) Characteristics of health professionals responsive to environmental health literacy Knowledge – Skills – Attitudes, (6) Interventions to improve environmental health literacy competencies.

## Data analysis

The extracted data were subjected to thematic analysis, following the methodological framework proposed by Nowell et al. [15]. This systematic approach allowed for the identification and organization of key patterns within the data, ensuring analytical rigor and transparency. The process involved six sequential steps: (1) Familiarization with the data through repeated reading and immersion in the content; (2) Generation of preliminary codes based on recurring elements and relevant features; (3) Identification of emergent categories and mapping of related competencies; (4) Review and refinement of categories to ensure internal coherence and distinctiveness; (5) Definition and naming of finalized thematic categories, reflecting core dimensions of environmental health literacy competencies; and (6) Construction of a comprehensive narrative that synthesizes the findings using the defined categories.

This analytical strategy enabled the development of a structured interpretation of the data, grounded in both empirical evidence and the conceptual aims of the review.

## Results

A comprehensive search was conducted across seven electronic databases (PubMed, Web of Science, Embase, Scopus, CINAHL, PsycINFO, and LILACS), yielding 3,842 records. In addition, 228 records were identified through other sources, including Google Scholar, OpenGrey, and ProQuest Dissertations & Theses.

After removing 308 duplicates, 3,534 unique references were screened by title and abstract. During the initial screening phase, 3,392 records were excluded for not meeting the scope of the review. Overall, 3,534 records underwent title and abstract screening, from which 142 studies were selected for full-text assessment. Of these, 117 articles were excluded for various reasons, including editorial or non-empirical publication type (n = 22), ineligible populations (n = 35), narrative review design (n = 10), and unavailability of the full text (n = 26).

Regarding records identified through additional search methods, 226 studies were screened, of which 222 were excluded due to population mismatch (n = 33) or lack of relevance to environmental health literacy (n = 189). At the end of the selection process, 29 studies met all eligibility criteria, comprising 25 studies identified through database searches and 4 studies retrieved through other search methods, including Google Scholar. The complete identification, screening, and selection process is presented in the PRISMA flow diagram (Fig 1).

Of the 29 included studies, five were theoretical contributions, comprising four literature reviews and one conceptual paper addressing curricular integration. These theoretical studies were used to inform the identification of core concepts and competencies related to environmental health, but were not included in the quantitative distributions of populations, settings, or geographical scope. The remaining 24 studies constituted the empirical evidence base for the analyses presented below and involved a broad range of health professional categories.

Among the empirical studies, most focused on practicing health professionals (n = 13), including nurses in diverse roles, midwives, perinatal health professionals, family physicians, pediatricians, public health personnel, community health

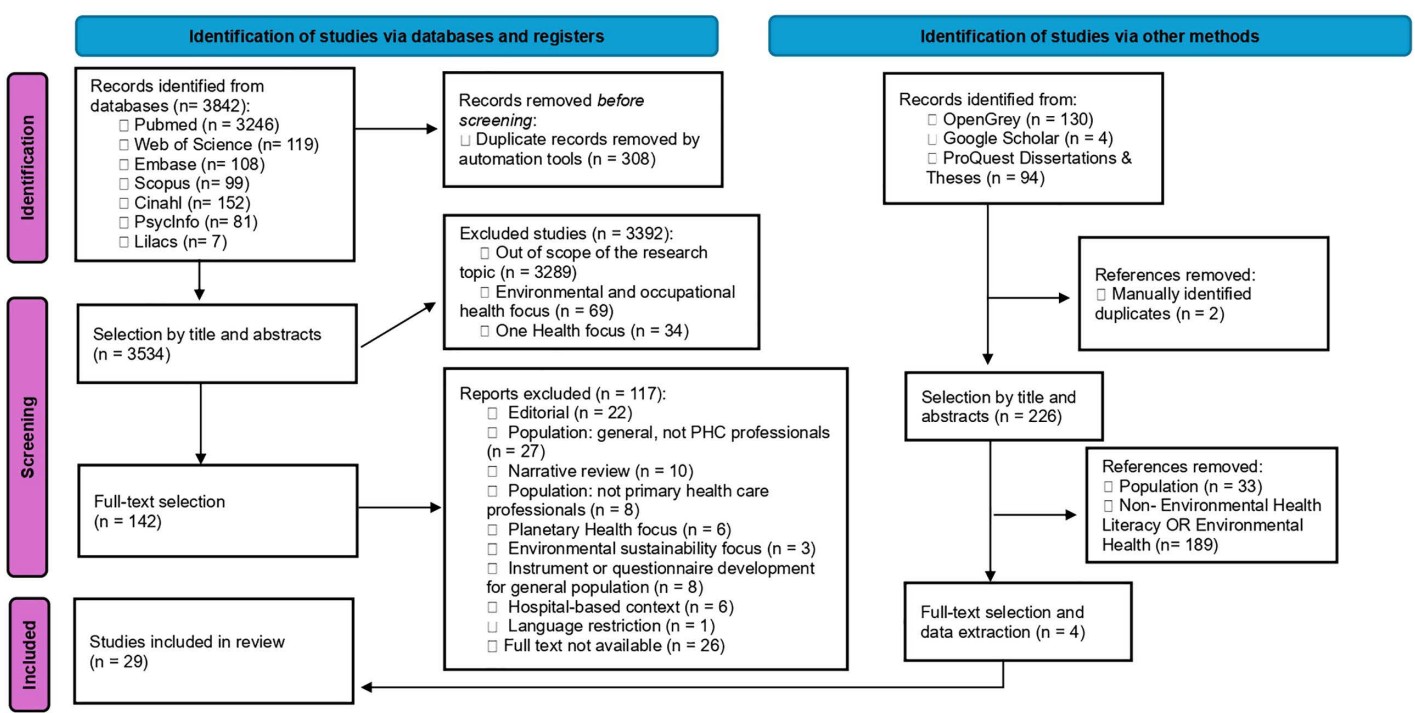

**Fig 1. PRISMA diagram – Selection of studies.**

workers, and multidisciplinary teams working in primary care or environmental health settings. Studies involving undergraduate students in nursing or medicine accounted for six studies (n = 6), reflecting a growing interest in introducing environmental health content during early professional training. In addition, two empirical studies (n = 2) specifically examined nursing faculty in relation to curricular integration and teaching practices. Three studies (n = 3) adopted a mixed population design, simultaneously including students and practicing health professionals, thereby bridging educational preparation and professional practice.

With respect to setting, and considering only the empirical studies, most investigations (n = 17; 58.6%) addressed Primary Health Care (PHC) directly, involving professionals embedded in PHC services such as family physicians, midwives working in PHC, community nurses, public health nurses, maternal and child health personnel, community health workers, perinatal care providers, and primary care teams. Only two empirical studies (n = 2; 6.9%) examined PHC indirectly, primarily through educational or conceptual approaches whose scope interfaces with PHC but is not explicitly situated within PHC service delivery. The remaining 10 empirical studies (34.5%) showed no operational or educational linkage to PHC, focusing instead on psychometric instrument development or theoretical propositions without explicit contextualization in primary care systems.

Geographically, and again based on the empirical studies, the evidence base was relatively concentrated. Most empirical work (n = 24) was conducted in the United States (n = 13; 54.1%), highlighting its leading role in environmental health and professional education research. Additional clusters of activity were observed in Spain (n = 4; 16.6%) and France (n = 2; 8.3%). Single empirical studies were identified in Turkey, South Africa, China, and Canada (each n = 1; 4.1%), contributing to the broader international scope of the field. One study (n = 1; 4.1%) represented an international collaboration between Spain and the United Kingdom. All included studies were published in English, consistent with the predominance of English-language scientific dissemination in environmental health research.

### Theoretical and conceptual foundations of environmental health literacy and its integration into primary health care education

Among the 29 studies included, only three explicitly addressed EHL as a core construct, using the term and defining it as a central analytical framework. In contrast, the remaining 26 studies (89.7%) engaged with EHL indirectly, examining environmental health–related knowledge, skills, competencies, practices, and educational strategies without explicitly labeling these elements as EHL.

This indirect engagement was reflected in the conceptual foundations and competency frameworks adopted across the studies. A majority of investigations (62.1%, n = 18) drew upon foundational definitions of environmental health, emphasizing exposures related to air, water, food, housing, and occupational settings as central components of PHC. Furthermore, 17 studies (58.6%) addressed the nature or levels of EHL, describing developmental progressions from basic awareness of environmental risks to more advanced competencies, including risk communication, population-level interventions, and policy advocacy. In parallel, 15 studies (51.7%) explicitly referenced core public health competencies, such as analytic and assessment skills, communication, cultural competence, community engagement, and leadership, thereby reinforcing the alignment between environmental health education and the operational requirements of PHC practice (Table 1).

Clinical environmental competencies, including environmental exposure history-taking and risk communication, were identified in 12 studies (41.4%), while a focus on vulnerable populations (particularly children, pregnant women, and environmentally burdened communities) appeared in 14 studies (48.3%). Overall, the distribution demonstrates strong conceptual convergence: across the literature, EHL is consistently framed as a multifaceted competency central to PHC practice, encompassing environmental risk assessment, clinical decision-making, community engagement, and policy-oriented action.

**Table 1. Conceptual and theoretical Foundations for Integrating Environmental Health Literacy into Primary Health Care (PHC) education (n = 29).**

| Theoretical and Conceptual Foundation | Key Elements and Components | Guidance for Integrating EHL into Educational Frameworks | n |
|---|---|---|---|
| Definition and Relevance of Environmental Health (EH) | Environmental Health is defined as the "freedom from disease or injury associated with exposure to toxic agents and other environmental conditions." Its scope encompasses air, water, soil, food, lifestyle factors, and occupational settings [17–19] | • Across-the-Curriculum Approach: EH content should be integrated horizontally across undergraduate and graduate curricula rather than taught as a separate subspecialty [20,21]<br>• Longitudinal Integration: EH should be introduced early and reinforced throughout the entire continuum of professional education [22]<br>• Leveraging Foundational Courses: Core Environmental Health content, such as toxicology and the principles of industrial hygiene, should be incorporated into existing foundational courses, including Advanced Health Assessment, Pathophysiology, and Pharmacology [17]<br>• Competency-Driven Integration: Curricular incorporation of Environmental Health is guided by competency-based learning objectives that structure both content and expected student outcomes [18,23] | 7 |
| The Nature of Environmental Health Literacy (EHL) | Environmental Health Literacy equips professionals to engage in preventive, upstream-oriented practice that addresses the population-level determinants of health, requiring the ability to read, interpret, and act upon environmental information [3,24]. Its development is conceptualized as a multilevel process [3,23,25–28]. | Levels of Environmental Health Literacy Integration in Professional Training: Level 1 – Foundational: Knowledge and understanding of environmental determinants and their effects on human health; Level 2 – Clinical: Ability to communicate environmental health risks and provide relevant guidance to patients; Level 3 – Community: Application of public health and population-based approaches to reduce exposures at the group level, including recognition of the distinction between individual and social determinants of health; Level 4 – Policy: Engagement in professional societies and policy arenas, participating in advocacy efforts through professional associations; Level 5 – Public Advocacy: Active involvement in public advocacy, influencing policy development and supporting legislative reforms aimed at protecting health [3]. | 7 |
| Essential Competencies | Environmental Health Literacy (EHL) reflects the ability to apply these responsibilities in routine practice—*how* professionals meaningfully engage with environmental health tasks. In this sense, EHL represents the demonstrable attainment of these essential competencies [3,20] | • Environmental Health Competencies: I) Fundamental Knowledge and Concepts; II) Environmental Exposure Assessment and Referral; III) Advocacy, Ethics, and Risk Communication; and IV) Legal and Regulatory Frameworks [17,18,20,22,24,25,29]<br>• Public Health Competencies: I) Analytic/Assessment Skills; II. Basic Public Health Sciences Skills; III. Cultural Competency Skills; IV) Communications Skills; V) Community Dimensions of Practice Skills; VI) Financial Planning and Management Skills; VII) Leadership and Systems Thinking Skills; e VIII) Policy Development/Program Planning Skills [28] | 9 |
| Core Clinical Competency | Understanding environmental exposures is fundamental to the training of health professionals and constitutes a core clinical competency within primary care practice [22,25]. | Training efforts should strengthen practical Environmental Health (EH) skills, with particular emphasis on the Environmental Exposure History (EEH) as a central clinical tool [22,30,31]. The **I PREPARE** mnemonic provides a structured approach for primary care providers to investigate environmental exposures by guiding questions about work, residence, environmental concerns, and daily activities, along with referrals and educatio [32]. The **CH$_2$OPD$_2$** mnemonic offers an additional framework for obtaining a comprehensive environmental exposure history, focusing on community, home, occupation [23] | 6 |
| Focus on Vulnerable Populations | EHL in the context of vulnerable populations is fundamentally linked to understanding both biological and social susceptibilities, as well as to the ability of health professionals to implement targeted interventions. [5,25,27,33–37] | Education aimed at preparing professionals to work with vulnerable groups must be multidisciplinary, practice-oriented, and culturally responsive. Key elements emphasized in the literature include: (I) Recognition of Unique Vulnerability [36,38]; (II) Understanding Biological Mechanisms [35,39]; (III) Identification of the "Risk Cocktail" [36,38]; (IV) Awareness of Socioeconomic Context [35,40] and (V) Perinatal and Preconception Exposure [36,40,41] | 12 |

## The specific knowledge, skills, and attitudes (competencies) that constitute EHL for PHC practice

Analysis of the 29 included studies identified five core competency domains that define EHL in PHC: knowledge, clinical skills, communication and counseling, systems-based practice and advocacy, and attitudes and equity. These domains were derived from the distribution of studies, the self-evaluation indicators proposed by the authors, and the representative competencies described in the literature (Table 2). Together, they reflect the multidimensional nature of EHL in PHC, integrating cognitive, clinical, communicative, systemic, and equity-oriented capacities.

Overall, knowledge emerged as the dominant domain (n = 13; 44.8%), followed by clinical skills (n = 7; 24.1%), communication and counseling (n = 6; 20.7%), attitudes and equity (n = 6; 20.7%), and systems-based practice and advocacy (n = 4; 13.8%). This distribution demonstrates that while foundational knowledge is well represented, competencies related to systemic navigation and advocacy appear less frequently in the literature.

The knowledge domain (44.8%) was the most frequently addressed. Self-assessment items emphasized preparedness to identify environmental determinants of health, understand pediatric vulnerability, and interpret basic exposure data. Correspondingly, representative competencies included understanding exposure pathways, recognizing community-relevant contaminants (e.g., endocrine disruptors), and keeping up with emerging environmental health risks.

Building on this foundation, studies addressing clinical skills (24.1%) focused on the practical application of knowledge in PHC settings. Self-evaluation items reflected confidence in conducting detailed occupational and residential exposure histories and using structured tools for environmental assessment. Representative competencies included differentiating clinically significant exposures and interpreting basic environmental indicators during patient encounters.

Complementing these skills, the communication and counseling domain (20.7%) highlighted providers' ability to translate complex environmental information into actionable guidance for patients. Self-assessment items centered on using plain language, offering preventive advice, and engaging patients in shared decision-making. Representative competencies included effective risk communication, patient education on exposure reduction, and contextualizing environmental data within clinical care.

The systems-based practice and advocacy domain (13.8%), though less frequently represented, emphasized competencies necessary for engaging with environmental health systems. Self-evaluation items covered the ability to identify appropriate referral resources and collaborate with public health agencies. Representative competencies included making referrals to environmental health services and advocating for community-level environmental protections.

Finally, the attitudes and equity domain (20.7%) underscored the ethical and justice-oriented dimensions of EHL. Self-evaluation items reflected the belief that environmental assessment should be a routine component of PHC practice and acknowledged the unequal distribution of environmental risks among socially vulnerable populations. Representative competencies included recognizing environmental health disparities, partnering with communities, and maintaining a proactive approach to environmental risk mitigation.

## Educational intervention outcomes in environmental health literacy

A total of 29 studies met the inclusion criteria for this scoping review. Among these, 14 studies implemented some type of educational intervention in Environmental Health (EH) or Environmental Health Literacy (EHL). However, not all reported quantitative outcomes or provided sufficient methodological detail to allow systematic extraction. Two studies—one describing only a curricular proposal [1] and another presenting interventions as part of narrative conclusions rather than as primary evaluative data [3] —were therefore excluded from the summary table. Table 3 focuses exclusively on the 12 studies that explicitly described an intervention and reported evaluable outcomes, either quantitative or qualitative (Table 3).

Across the 12 included intervention studies, sample sizes ranged from 8 to 994 participants, encompassing undergraduate and graduate students, midwives, nurses, physicians, public health professionals, and faculty members. The interventions employed a wide range of pedagogical formats, with the most effective approaches consistently combining traditional didactic instruction with active, immersive, and technology-mediated methods. These strategies aimed not only

**Table 2.** Core Competency Domains for Environmental Health Literacy in Primary Health Care (PHC) (n = 29).

| Domain | Suggested Self-Assessment Indicators of Competence | Representative Competencies | n |
|---|---|---|---|
| Knowledge | • I feel adequately prepared to identify the influence of environmental factors and environmental agents on the health of my patients.<br>• I possess sufficient knowledge regarding the unique biological and behavioral vulnerability of pediatric populations to environmental exposures.<br>• I feel confident in analyzing and discussing knowledge gaps related to endocrine disruptors and other relevant contaminants present in the community where I practice.<br>• My current knowledge allows me to interpret basic environmental exposure data (e.g., air or water quality reports). | • Understand the influence of environmental factors and environmental agents on human health [17,18,22,25,29,42]<br>• Recognize the unique vulnerability of pediatric populations, given their increased exposure and biological immaturity [17,27,31,39,43]<br>• Analyze knowledge gaps in more complex domains—such as biomarkers, pesticides, and particularly endocrine disruptors—present within the community in which the health professional practices [18,26,40,41] | 13 |
| Clinical skills | • I routinely obtain a detailed environmental exposure history (including occupational and residential factors) for all of my patients.<br>• I feel capable of using standardized tools or mnemonics (e.g., $CH_2OPD_2$) to guide the collection of environmental exposure information.<br>• I am able to distinguish clinically relevant environmental exposures from low-risk exposures during patient assessment. | • Conduct environmental exposure histories and interpret basic exposure data [3,22,23,25,29]<br>• Obtain a detailed environmental exposure history—including occupational and residential factors—from all patients [22,30,31] | 7 |
| COMMUNICATION AND COUNSELING | • I consistently use plain language and avoid technical jargon when discussing environmental risks with my patients.<br>• I actively provide health education to my patients on strategies to reduce exposure to contaminants in food and household products.<br>• I employ shared decision-making techniques to ensure that patients actively participate in selecting prevention and risk-reduction strategies. | • Use plain language, risk communication strategies, and shared decision-making techniques [25]<br>• Demonstrate the ability to discuss environmental risks and provide patients with clear, comprehensible information on risk-reduction and prevention strategies [5,18,22,25,38,43] | 6 |
| Systems and advocacy | • I am able to identify and refer patients to appropriate sources of information and relevant public health regulatory agencies for environmental health concerns.<br>• I engage in advocacy activities (e.g., notifying public authorities, participating in committees) aimed at reducing the impact of pollutants in my community.<br>• I feel responsible for alerting public agencies about Environmental Health hazards that I identify in my clinical practice. | • Identify relevant reference sources and engage with regulatory and public health agencies [18,22]<br>• Demonstrate competence in advocating within the community to reduce pollutant impacts and in notifying public authorities about environmental health hazards [18,19,22,36] | 4 |
| Attitudes and equity | • I feel effective in applying risk communication strategies to help patients understand the probability and severity of an exposure.<br>• My current training has provided me with the knowledge and skills necessary to counsel patients on strategies for reducing environmental risk.<br>• Environmental Health assessment should be a mandatory and systematic component of every clinical encounter, even in the absence of obvious symptoms.<br>• Lack of time during clinical visits is a justifiable barrier to conducting a detailed environmental exposure assessment. *(Reverse-coded statement)*<br>• I recognize that exposure to environmental hazards is unequal and disproportionately affects socioeconomically vulnerable populations in the area where I practice.<br>• I actively seek to engage patients and the broader community as partners in identifying and addressing environmental health problems. | • Recognize environmental health disparities and adopt a community-partnership orientation [35]<br>• Affirm that Environmental Health assessment should be part of routine clinical practice [18,25,41]<br>• Demonstrate perceived self-competence to address environmental exposures [24,25,40] | 6 |

**Table 3. Environmental Health Literacy Intervention Studies for Health Students and Professionals (n = 12).**

| Study | Context and Population | Type of Intervention | Methods | Competencies Developed (K/S/A)* | Intervention Outcomes |
|---|---|---|---|---|---|
| [36] | Continuing Education (CE) Sample: 42 midwives/nurses in primary health care services | Short-term in-person training Duration: 2 days Focus: Environmental and child health | Quasi-experimental (single-group pretest–posttest) | **K:** Knowledge of environmental and child health, risk factors **S:** Ability to identify at-risk children and analyze exposure history **A:** Increased awareness | Significant improvement in knowledge (pre: 20.29; post: 31.07; $p < 0.001$). Large effect size (Eta² = 0.78). |
| [35] | Undergraduate/Graduate Programs Sample: 116 nursing students and nurses | Hands-on modeling, case study, and in-person lectures Duration: 3 hours Focus: Environmental Health Literacy and Genomics | Survey with pretest–posttest design | **K:** Molecular basis, toxicology **S:** Ability to communicate concepts using culturally appropriate language **A:** Increased confidence and literacy in environmental health and genomics | Significant knowledge gain ($p < 0.0001$). Mean scores increased from 10.9 to 14.0 (+3.1). |
| [18] | Undergraduate Programs Sample: 36 medical students | Six-week module integrating interactive classes and use of a product-evaluation app Focus: Environmental Health | Survey-based study | **K:** Knowledge on toxins/endocrine disruptors **S:** Counseling skills, community advocacy **A:** Perceived preparedness/self-efficacy | Likert-scale increases: • Diet/toxins counseling: 2.31→3.46 • Personal care/household products: 2.48→3.47 • Community advocacy: 2.27→3.39 (all $p < 0.001$) |
| [44] | Continuing Education (CE) Sample: 994 participants (nurses, mid-level practitioners, medical/nursing students, and other non-clinical professionals) | Digital training with 12 online modules Duration: 30–60 minutes per module Focus: Pediatric Environmental Health | Intervention study with follow-up survey | **K:** Increased knowledge of pediatric environmental health **S:** Implicit increase in skills | Mean score improved from 62.33% to 93.07% (+30.74%; $p < 0.0001$). Gains by subgroup: nurses +33.32%, mid-level +33.24%, physicians +28.58%, students +30.04%. ANOVA: $p = 0.01$. |
| [38] | Undergraduate Programs Sample: 267 nursing students (110 Spain; 157 UK) | BOLD (blended learning) model combining e-learning and case-based learning Duration: 90 minutes Focus: Pediatric Environmental Health | Quasi-experimental time-series design | **K:** Knowledge on child environmental health and climate change **S:** Problem-solving, health education on contaminants **A:** Attitudes, commitment, agency as change agents | Significant improvements ($p < 0.001$): • Knowledge +39.02% • Skills +29.98% • Attitudes +15.81% "Good knowledge": Spain 39.09%→90%; UK 19.11%→64.33%. |
| [40] | Continuing Education (CE) Sample: 11 participants (midwives, nursery nurse, GP, obstetrician) | Training day (Photolanguage 1h30; lecture 1h; 62-min focus group) Focus: Environmental Health | Qualitative (focus group) | **K:** Relationship between social environment and patient behavior **S:** Development of culturally relevant educational strategies **A:** Increased sensitization to environmental health issues | Improved attitudes, communication, and awareness of modern environmental risks among vulnerable populations. |
| [5] | Continuing Education (CE) **Sample:** ~86 health professionals (physicians, nurses, public health workers) | Development and implementation of the multimedia e-book A Story of Health Focus: Environmental Health Literacy | Descriptive study | **K:** Interaction of environmental and genetic factors **S:** Ability to develop strategies and interventions **A:** Environmental health literacy; recognition of professionals as trusted messengers | 90% user satisfaction, applicability to practice, strategy development, and perceived absence of barriers. |
| [31] | Continuing Education (CE) **Sample:** Nursing researchers, public health nurses, environmental health specialists, research coordinators | Active in-person methodology (home-based intervention) **Focus:** Environmental Health | Case study | **K:** Child vulnerability and household exposures **S:** Ability to conduct full home environmental risk assessments, provide education and referrals | Not applicable (descriptive case). |

*(Continued)*

**Table 3.** (Continued)

| Study | Context and Population | Type of Intervention | Methods | Competencies Developed (K/S/A)* | Intervention Outcomes |
|---|---|---|---|---|---|
| [17] | Graduate Programs **Sample:** Master's students (Family Nurse Practitioner – FNP e Clinical Nurse Specialist – CNS) | 3-hour lecture and discussion **Focus:** Environmental Health | Descriptive study | **K:** Toxicology/industrial hygiene principles **S:** Application of the Hierarchy of Controls **A:** Increased acceptance of environmental health content | Environmental exposures documented in only 9.4% of 53 audited records; Hierarchy of Controls identified as the most difficult concept. |
| [29] | Continuing Education (CE) **Sample:** Nursing faculty and public health nurses | 2.5-day community case study conference; 6-hour interpreted bus/walking tour **Focus:** Environmental Health | Descriptive study | **K:** Toxicology, epidemiology **S:** Environmental exposure history, community assessment, digital resource use **A:** Social justice awareness, literacy and sensitization | 100% of participants rated the combined lecture–hands-on–field visit method as highly effective. |
| [21] | Undergraduate Programs **Sample:** 94 nursing students in a community health nursing course | Integration of in-person/online classes with 3–8 hour fieldwork supervised by environmental health professionals **Focus:** Environmental Health | Intervention study | **K:** Environmental health subspecialties **S:** Development of environmental risk assessments during home visits **A:** Increased valuation of environmental health knowledge, enhanced worldview | 73.4% positive evaluation; 12.8% found it valuable despite limited initial interest. |
| [20] | Continuing Education (CE) **Sample:** Clinical nurses, academics, researchers | Five-day conference/lecture **Focus:** Environmental Health | Nominal Group Process (NGP) | **K:** Basic mechanisms of environmental exposures, prevention and control strategies **S:** Comprehensive environmental exposure history, appropriate referrals **A:** Adoption of the role of patient advocate | Priority strategies identified: make Environmental Health mandatory in curricula; strengthen interprofessional collaboration; offer placements in agencies; train faculty; create Environmental Health awards. |

**Notes**: *Developed Competencies (Knowledge – K, Skills/Practice – S, Attitudes/Awareness – A).

to improve knowledge but also to enhance self-efficacy, a key determinant of preparedness to recognize environmental exposures and provide counseling. Knowledge was addressed in 100% of studies, followed by skills (92%) and attitudes/awareness (83%). In formal educational settings, the predominant curricular strategy was the integration of EH/EHL content into existing courses, rather than the creation of new standalone modules, reflecting the structural constraints of already saturated curricula.

Educational outcomes were consistently positive. Short-term in-person training for midwives increased mean knowledge from 20.29 to 31.07 (+53%, $p < 0.001$). Hands-on modeling for undergraduate nursing students (n = 116) increased test performance from 10.9 to 14.0 (+3.1 points; $p < 0.0001$). A six-week blended module for medical students (n = 36) produced substantial gains in self-reported preparedness, with improvements of 33–52% across counseling domains ($p < 0.001$). Large-scale digital training through the PEHSU program (n = 994) raised mean scores from 62.33% to 93.07% (+30.74%; $p < 0.0001$). A blended intervention with nursing students (n = 267) increased knowledge by 39.02%, skills by 29.98%, and attitudes by 15.81% ($p < 0.001$), with "good knowledge" rising from 39.09% to 90% in Spain and 19.11% to 64.33% in the UK.

Experiential and community-based approaches also yielded strong outcomes. A 2.5-day case-based conference reached 100% agreement among participants regarding the effectiveness of combining lectures, laboratory demonstrations, and a 6-hour field tour to make complex toxicology and environmental epidemiology concepts accessible. In

undergraduate field placements, 73.4% of students (69/94) evaluated the intervention positively, while 12.8% acknowledged its value despite low initial interest.

Process and behavioral indicators revealed areas for improvement. Among postgraduate trainees, only 9.4% of clinical records (5/53) documented environmental exposures, and the Hierarchy of Controls was identified as the most challenging concept. Taken together, these findings indicate that although multimodal and blended educational strategies significantly enhance EHL knowledge, skills, and self-efficacy, sustained curricular integration and structured practical application remain essential for translating competencies into clinical and community practice.

## Discussion

The findings of this scoping review reinforce the understanding of EHL as a multidimensional construct that extends beyond the mere acquisition of factual knowledge. EHL consistently emerged as a layered developmental process in which professionals progress from basic awareness of environmental determinants of health toward more advanced competencies involving clinical reasoning, community engagement, and policy-oriented action [45]. This conceptual progression mirrors established models of professional development in PHC, where practitioners evolve from applying individual-level interventions to engaging in upstream strategies that address structural and environmental determinants [46]. The recurrent emphasis on pediatric populations, pregnant women, and socioeconomically vulnerable groups underscores the ethical imperative inherent in EHL: the need to safeguard communities that disproportionately bear the burden of environmental exposures. This equity-focused orientation provides a unifying conceptual thread across diverse educational and clinical contexts [47].

The identification of five interrelated competency domains, knowledge, clinical skills, communication and counseling, attitudes and equity, and systems-based practice and advocacy, offers a coherent and comprehensive framework for conceptualizing EHL in PHC settings. However, the predominance of knowledge-centered outcomes relative to systems-based and advocacy-oriented competencies reveals a persistent imbalance in current educational approaches. While foundational knowledge is essential for recognizing environmental hazards, effective PHC practice depends on the ability to navigate regulatory systems, mobilize community resources, and influence upstream determinants [48]. Concurrently, the limited attention to communication and counseling competencies is concerning, given the centrality of risk communication in PHC and the complexities involved in helping patients interpret and act upon environmental health information. Collectively, these gaps suggest that existing curricula may enhance understanding without fully preparing professionals for practical, context-specific application, or for the intersectoral action that is fundamental to an integral approach to environmental health [49].

In this context, the educational interventions identified in the review provide an important counterpoint. They demonstrate considerable promise for strengthening EHL, with substantial gains observed across knowledge, skills, and attitudes. The consistent effectiveness of blended and technology-enhanced approaches indicates that pedagogical diversity, particularly when incorporating experiential components, can meaningfully enhance learners' self-efficacy, a key determinant in translating knowledge into action [50]. Nevertheless, the concentration of interventions among undergraduate students and nurses, alongside the limited participation of postgraduate trainees and broader PHC teams, underscores that these gains have not yet been equitably distributed across the workforce. Expanding educational initiatives to involve a wider range of PHC professions could help build a more cohesive and system-responsive workforce, capable of integrating EHL more effectively [51].

Despite these encouraging developments, the translation of EHL gains into clinical practice remains limited. The strikingly low documentation of environmental exposures in clinical records exemplifies a persistent implementation gap in EHL, underscoring that improved knowledge alone does not guarantee practice change. Structural barriers, including curricular overload, limited consultation time, lack of incentives, and insufficient organizational support, emerged consistently across studies, indicating that individual-level training is necessary but not sufficient [52]. These findings highlight

the importance of shifting from isolated educational strategies to comprehensive implementation approaches that address workflow integration, institutional infrastructure, and policy-level enablers. Without such systemic alignment, EHL risks remaining an aspirational competency rather than an operationalized component of PHC [53].

This structural pattern is further reflected in the geographic and professional distribution of the EHL evidence. The predominance of studies conducted in high-income countries suggests that prevailing EHL frameworks may insufficiently capture the realities of low- and middle-income settings, where environmental health burdens are often more acute [54]. Similarly, while the central role of nurses in EHL research aligns with their preventive and community-based orientation, the relative absence of other PHC professionals points to underutilized opportunities for broader interprofessional engagement. Addressing these gaps will be essential to advance EHL as a shared PHC responsibility rather than a niche domain concentrated within a single professional group.

Against this backdrop, several implications arise for research, policy, and educational planning. From a research standpoint, there is an urgent need for implementation science approaches to examine how organizational structures, clinical workflows, and institutional environments facilitate or impede the adoption of EHL competencies [55]. Additional inquiry in diverse global contexts would strengthen the applicability of existing frameworks and ensure that emerging initiatives align with the environmental health challenges faced by different health systems. From a policy perspective, sustainable EHL integration will require coordinated action across educational institutions, accreditation bodies, and healthcare organizations. Embedding EHL competencies into professional training standards, developing supportive documentation and referral systems, and creating institutional incentives for environmental health assessment represent promising strategies for fostering durable adoption. Furthermore, the development and validation of standardized EHL assessment tools could enable consistent measurement of educational and practice outcomes.

The limitations of the assessed evidence base should temper interpretation. The heterogeneity of populations, interventions, and outcomes precluded meta-analysis and limited direct comparisons of effectiveness. The predominance of studies from high-income countries may also restrict generalizability to regions where environmental burdens are more severe and PHC systems are more resource constrained. Even so, the evidence converges on a clear conclusion: EHL is essential for contemporary PHC, educational interventions can substantially improve competencies, and meaningful public health impact will depend on coordinated action across educational systems, health service organizations, and policy environments.

## Conclusion

This review establishes EHL as a foundational and multidimensional competency within community-based primary health care, intersecting preventive, clinical, communicative, and advocacy capacities. Although conceptual heterogeneity exists in the literature, five core domains consistently define effective EHL: environmental health knowledge, clinical exposure assessment, risk communication, equity-oriented attitudes, and systems-level advocacy. Multimodal training strategies, especially those enhanced by digital tools and mobile learning systems, were associated with meaningful gains in knowledge, skills, and professional self-efficacy among PHC teams.

However, evidence of translation into routine practice remains limited, revealing a persistent implementation gap. Inconsistent recording of environmental exposures in standardized medical registries, lack of structured reporting instruments, and weak institutional integration into daily PHC workflows indicate that education, while essential, cannot sustain adoption without enabling infrastructures. Bridging this disconnect will require embedding EHL into undergraduate and residency medical training pathways, aligned with competency-based assessment models, institutional protocols, and interoperable digital health platforms that support longitudinal exposure documentation and interprofessional collaboration.

Strengthening environmental health capacity in PHC demands a coordinated transformation spanning academic institutions, service management systems, public policy frameworks, and environmental-health partnerships. Systemic

alignment, measurable indicators, and routine documentation standards will be critical to ensure meaningful clinical and public-health impact.

## Supporting information

**S1 File. Appendix 1.**
(DOCX)

## Author contributions

**Conceptualization:** Flaviane Cristina Rocha Cesar, Fabian Calixto Fraiz, Aline Maciel Monteiro.

**Formal analysis:** Flaviane Cristina Rocha Cesar, Fabian Calixto Fraiz, Aline Maciel Monteiro.

**Writing – original draft:** Flaviane Cristina Rocha Cesar, Fabian Calixto Fraiz, Aline Maciel Monteiro, Maria Alves Barbosa, Andréa Cristina de Sousa, João Victor Pereira Almeida, Jaqueline Reis Sotero da Silva.

**Writing – review & editing:** Flaviane Cristina Rocha Cesar, Fabian Calixto Fraiz, Aline Maciel Monteiro, Maria Alves Barbosa, Ricardo Cambraia Parreira, Andréa Cristina de Sousa, João Victor Pereira Almeida, Jaqueline Reis Sotero da Silva.

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
