## [Decision Letter · Decision Letter 0]

2 Feb 2026

Environmental Health Literacy Competencies and Teaching Methods for Students and Health Professionals in Primary Health Care: A Scoping Review

PONE-D-25-64280

Dear Dr. Cesar,

We’re pleased to inform you that your manuscript has been judged scientifically suitable for publication and will be formally accepted for publication once it meets all outstanding technical requirements.

Kind regards,

André Luis C Ramalho, PhD

Academic Editor

PLOS One

Reviewers' comments:

Reviewer's Responses to Questions

**Comments to the Author**

1. Is the manuscript technically sound, and do the data support the conclusions?

Reviewer #1: Yes

Reviewer #2: Yes

2. Has the statistical analysis been performed appropriately and rigorously?

Reviewer #1: N/A

Reviewer #2: Yes

3. Have the authors made all data underlying the findings in their manuscript fully available?

Reviewer #1: Yes

Reviewer #2: Yes

4. Is the manuscript presented in an intelligible fashion and written in standard English?

Reviewer #1: Yes

Reviewer #2: Yes

**Reviewer #1:**  Dear Authors, Dear Authors,

- This topic is timely and well‑motivated given the expanding remit of PHC to address environmental determinants of health. The proposed EHL competency framework is a key strength. Besides, the manuscript is methodologically sound and consistent with PRISMA‑ScR and JBI guidance, and its conclusions are reasonably supported by the mapped evidence.

- No original statistical analysis was conducted by the authors, which is appropriate for a scoping review. Descriptive reporting of quantitative outcomes from included studies is adequate.

- The final queries applied in each database are shared as an Appendix of this article, which ensures the transparency of the applied methods, thus allowing a user to address the same question and screen the same set of literature to come up with a comparable general conclusion.

- The manuscript is generally clear, well structured, and written in acceptable scientific English.

Given all these aspects, this manuscript is recommended for publication.

**Reviewer #2:**  I have no comments to the author at this time. The article is well written and I have no additions at this time. The authors should keep up this approach to developing and crafting future articles. The attention to detail is emulable. I have no comments to the author at this time. The article is well written and I have no additions at this time. The authors should keep up this approach to developing and crafting future articles. The attention to detail is emulable.

**Do you want your identity to be public for this peer review?** For information about this choice, including consent withdrawal, please see our For information about this choice, including consent withdrawal, please see our Privacy Policy .

Reviewer #1: **Yes:** Bruno Filipe Coelho da CostaBruno Filipe Coelho da Costa

Reviewer #2: **Yes:** Henry J. lawsonHenry J. lawson

---

## [Editor Report · Acceptance letter]

PONE-D-25-64280

PLOS One

Dear Dr. Cesar,

I'm pleased to inform you that your manuscript has been deemed suitable for publication in PLOS One. Congratulations! Your manuscript is now being handed over to our production team.

Kind regards,

on behalf of

Prof. Dr. André Luis C Ramalho

Academic Editor

PLOS One